# Evaluation of Cytotoxic and Antibacterial Effect of Methanolic Extract of *Paeonia lactiflora*

**DOI:** 10.3390/medicina58091272

**Published:** 2022-09-14

**Authors:** Yu-Ri Choi, Min-Kyung Kang

**Affiliations:** 1Department of Dental Hygiene, Hallym Polytechnic University, Chuncheon 24210, Gangwon-do, Korea; 2Department of Dental Hygiene, Hanseo University, Seosan 31963, Chungcheongnam-do, Korea

**Keywords:** anti-bacterial, *Streptococcus mutans*, *Candida albicans*, *Paeonia lactiflora*

## Abstract

Background and objectives: Bacterial antibiotics have had several side effects. Therefore, interest in natural substances with less side effects is increasing these days. *Paeonia lactiflora*, the root of *Paeonia lactiflora*, is used as a raw material for medicines. In this study, we investigated the antibacterial effect and the cytotoxicity of *Paeonia lactiflora* extract. Materials and Methods: For cytotoxicity, MTT analysis according to ISO 10993-5 was performed. The antibacterial test of the *Paeonia lactiflora* was determined from bacterial viability, Inhibition zone test, CFU (colony forming unit) and SEM (scanning electron microscope). To confirm the antibacterial component of *Paeonia lactiflora*, the content of flavonoids and polyphenols was analyzed. Results: Our results showed that *Paeonia lactiflora* extract contained flavonoids and polyphenols, which exhibited antimicrobial activity against *Streptococcus mutans* (*S. mutans*) and *Candida albicans* (*C. ablicans*). Further, the cytotoxicity of *Paeonia lactiflora* extract was low. Conclusions: We believe that our study makes a significant contribution to the literature because it demonstrates that *Paeonia lactiflora* extract can be used as an antibiotic.

## 1. Introduction

The oral cavity is in contact with the environment outside the body, allowing microorganisms to grow nutritionally and physiologically. And a bacterial flora form in the oral cavity. It is known that there are more than 30 types of bacteria in the oral cavity of an average person, which varies depending on the individual, age, health condition and diet [1,2]. Normal oral microflora typically contains *Streptococcus mutans* (*S. mutans*), *Prevotellar intermedia* (*P. intermedia*), *Porphyromonas gingivalis* (*P. gingivalis*), *Actinobacillus actinomycetemcomitans* (*A. A*), *Candida albicans* (*C. albicans*) [3]. Among them, *S. mutans* causes dental caries, a disease caused by damage to the enamel of teeth by acid produced by the decomposition of sugar by bacteria in the dental microbiota [4]. In addition, candidiasis, which occurs in the fungus candida albicans, causes vaginal candidiasis, deep mucosal infections (esophagitis, vaginitis, and intestinal candidiasis), and causes thrush when the oral mucosa occurs [5]. *S. mutans* bacteria are resident bacteria in the oral cavity, but when the amount increases, it causes dental caries.

The most widely used antibacterial agents are phenolic compounds (triclosan), quaternary ammonium agents (benzalkonium, chloride), and Chlorohexidine [6,7]. These chemical agents cause many side effects such as toxicity, carcinogenesis, and mutagenesis [8]. So, previous research is being actively conducted to find excellent oral disease preventive substances from natural sources that are suitable for long term use because which have fewer side effects [9,10]. In addition, the antibacterial effect of natural extracts has been demonstrated in relation to antioxidants. Natural sulfates include plant-derived products, such as phenolic compounds, flavonoids, and vitamins [9]. Various natural extracts have been found to have antioxidant activity, such as propolis and green tea extracts [9,10,11,12,13].

The root of the *Paeonia lactiflora* (Paeoniae Radix Pall) is used as a raw material for various medicines [14]. Physiological activity of the *Paeonia lactiflora* has been researched for its anti-allergic, analgesic, antibacterial and whitening effects. Paeoniflorin, a major constituent in herbal medicine, is reported to have various activities such as immune-related activity, anti-inflammatory, and anti-allergic activity [15,16]. In this study, the antibacterial effect and the cytotoxicity of *Paeonia lactiflora* extract were confirmed and used as basic data to support the utilization of natural extract materials. The hypothesis of our thesis is that the *Paeonia lactiflora* extract will have an antibacterial effect against *S. mutans* and *C. albicans*.

## 2. Materials and Methods

### 2.1. Material

*Paeonia lactiflora* extract was purchased from *Paeonia lactiflora* (Jirisangol, Sancheong-gun, Gyeongsangnam-do, Korea) 500 g was crushed and extracted in 70% methanol solution 10 times by weight at room temperature for 48 h.

After filtering the extract with filter paper (Filter paper #2, Whatman, Maidstone, Kent, UK), the filtered extract was concentrated by evaporation using a vacuum evaporator (EYELA, Tokyo, Japan). Application of the experimental group of the concentrated extract was dissolved in ethanol at 50 μg/mL, 100 μg/mL, 150 μg/mL, and 200 μg/mL. All experimental groups were repeated 10 times.

### 2.2. Content in Paeonia lactiflora

#### 2.2.1. Content of Flavonoid

First, 1 mL diethylene glycol was added to 100 µL of the test solution, and 100 µL of 1 N and NaOH was mixed. Then, after reacting in a constant temperature water bath at 37 °C for 1 h, absorbance was measured at 420 nm using a UV Spectrometer. A standard curve was drawn using the standard naringin (Sigma Aldrich, St. Louis, MO, USA) and the content of flavonoids was calculated. All experimental groups were repeated 10 times.

#### 2.2.2. Contents of Polyphenol

The content of polyphenols was measured using a diethylene glycol solution that was immersed in distilled water for 7 days in a constant temperature water bath at 37 °C. 650 µL of distilled water was added to 50 µL of the test solution. Then, 50 µL of folin denis reagent was added and reacted at room temperature for 3 min. Saturated 10% Na_2_CO_3_ solution was added to 100 µL, and 150 µL of distilled water was added and mixed to adjust the final 1 mL volume. Then, after reacting in a constant temperature water bath (darkroom) for 37 h, the absorbance was measured at 725 nm using a UV/VIS Spectrometer (Human, Yongin-si, Korea). A standard curve was drawn using the standard garlic acid (Sigma, Darmstadt, Germany) and the content of polyphenols was calculated.

### 2.3. Cytotoxicity Test

#### 2.3.1. Material for Cytotoxicity Test

The freeze-dried natural extract was placed in a mortar and pulverized into a powder form. The pulverized powder was added to RPMI 1640 (Gibco Laboratoties, Grand Island, NY, USA) to prepare concentrations of 200.0 μg/mL, 150.0 μg/mL, 100.0 μg/mL, and 50.0 μg/mL.

#### 2.3.2. MTT Assay

According to the MTT cytotoxicity test method of ISO 10993-5, the number of L929 cells per well was adjusted to 1 × 104, and 100 μL was dispensed into wells and cultured for 24 h. After incubation, 100 μL of the natural extract diluted to various concentrations was applied to the cells for 24 h. As a control, RPMI 1640 without natural extract was used. After application, the extract was discarded and washed with 100 μL of DPBS (Gibco BRL, Life Technologies, New York, NY, USA). After washing was completed, DPBS was removed, and 50 µL per well was added to the culture solution containing 1 mg/mL of MTT (Sigma, Darmstadt, Germany), followed by additional culture for 2 h. To dissolve the formed MTT (3-[4,5-Dimethylthiazol-2-yl]-2,5-diphenyltetrazolium bromide) formazan, 100 mL of isopropanol (Sigma, Darmstadt, Germany) was added to 100 µL/well and reacted for 20 min. After that, the absorbance was measured at 570 nm on a spectrophotometer and analyzed. The result was normalized to 100% of the MTT reduction rate of the control group and expressed as a percentage.

#### 2.3.3. Microscopic Observation

For cell morphology observation, the L929 cells exposed to various concentrations of natural extracts were observed through an EVOS FL microscope (Advanced Microscopy, Washington, DC, USA) at 20× magnification before application to the MTT solution.

### 2.4. Antibacterial Effect Test

#### 2.4.1. Microbial Preparation

The strains used in the experiment were *Streptococcus mutans* (ATCC 25175) and *Candida albicans* (ATCC 10231). *Streptococcus mutans* was in brain heart infusion media (Becton Dickinson and Co., Sparks, MD, USA). *Candida albicans* is yeast mold (Becton Dickinson and Co., Franklin Lakes, NJ, USA) inoculated in a liquid medium and incubated at 37 °C Incubated for 24 h and used in the experiment.

#### 2.4.2. Bacterial Viability (Optical Density)

To analyze the effect of *Paeonia lactiflora* extract on bacterial growth inhibition, a liquid medium dilution method was used. The specimen was immersed in 600 μL of PBS and eluted for 24 h. The bacterial culture was diluted with an OD(optical density) 600 value of 0.4–0.6. Absorbance was measured at 600 nm using an ELISA reader (Epoch, BioTeck, Winooski, VT, USA). The eluate and the bacterial culture were mixed 1:1 and cultured in a 37 °C incubator for 24 and 48 h.

#### 2.4.3. Inhibition Zone Test

An inhibition zone test was used to confirm the growth inhibition effect of bacteria. Bacteria were prepared in 100 ul and applied to each medium. BHI and YM media were made with bacterial suspension. The filter paper was manufactured in the form of a disc, 10 mm in diameter, and sterilized. The solution was prepared, and 20 μL was applied uniformly to the extract on a filter paper. The bacteria were prepared by applying them on a solid agar plate and wet in filter paper. And then Stored in an incubator at 37 degrees for 24 h. After 24 h, the size of the inhibition zone of each bacterium was measured using a Vernier caliper.

#### 2.4.4. CFU (Colony Forming Unit)

The sample solution and microbial culture (1 × 10^5^ cells/mL) was mixed at 1:1 ratio. Of this mixture, 100 μL was spread onto a BHI and YM agar plate and incubated at 37 °C for 24 h. Then, the total number of colonies was counted.

#### 2.4.5. SEM (Scanning Electron Microscope)

1 mL of microbial suspension (1 × 10^5^ cells/mL) was placed on a 24-well plate and incubated at 37 °C for 24 h. For microscopic examination, microbe was fixed with 2% glutaraldehyde-paraformaldehyde in 0.1 M PBS for at least 30 min, at room temperature. The samples were post-fixed with 1% OsO_4_ dissolved in 0.1 M PBS for 2 h dehydrated with gradual ethanol, treated with isoamyl acetate, and subjected to critical point drying (Leica, Wien, Austria). Then, the samples were coated with Pt (5 nm) by using an ion coater. They were then examined and photographed using a scanning electron microscopy (Carl Zeiss, Oberkochen, Germany) at 2 kV. SEM observed at a magnification of 20,000×.

## 3. Results

### 3.1. Content of Flavonoid and Polyphenol

The concentration of flavonoids and polyphenols increased with increasing the *Paeonia lactiflora* extract concentrations up to 150 μg/mL, except for 200 μg/mL (Table 1). And the contents of polyphenol and flavonoid of all experimental group higher than control groups of 0 μg/mL (*p* < 0.05).

### 3.2. Result of Cell Viability Test (MTT Assay)

The MTT assay results are shown in Figure 1. The cell viabilities of the 50 μg/mL, 100 μg/mL, 150 μg/mL, and 200 μg/mL experimental groups were 80.2%, 79.5%, 70.0%. All groups were more than 70%, respectively. Above 150 μg/mL groups, there was a significant difference compared to the below 100 μg/mL group (*p* < 0.05). However, the cell viability of all experimental group is high. In the similar trend to the MTT result, the morphology of the cells appeared in each experimental group, and at 200 μg/mL, the cell morphology appeared as a spindle shape. There were cells that appeared of live (Figure 2). Therefore, Cell shape was normal in all the experimental group.

### 3.3. Antibacterial Activity

#### 3.3.1. Viability of Bacteria (OD)

The optical density (OD) results are shown in Figure 3. The bacterial viability of both *S. mutans* and *C. albicans* was decreased in experimental groups containing *Paeonia lactiflora* extract compared to control group (*p* < 0.05). However, there were no significant difference among the experimental group (*p* > 0.05).

#### 3.3.2. The Size of the Inhibition Zone Result

In order to evaluate the antibacterial activity of the extract, the size of the experimental group was checked to confirm the size of the inhibition zone and the results are shown in Figure 4. The results on *S. mutans* showed that 1.07 ± 0.06 at 50 μg/mL, 100 μg/mL was 1.10 ± 0.00, 150 μg/mL was 1.13 ± 0.06, and the 200 μg/mL group was 1.27 ± 0.06. And size of inhibition zone on *C. albicans* were 1.20 ± 0.10, 1.33 ± 0.21, 1.40 ± 0.17, and 1.47 ± 0.15 respectively at concentrations of 50 μg/mL, 100 μg/mL, 150 μg/mL, and 200 μg/mL. The antimicrobial activity against *S. mutans* and *C. albicans* showed a significant difference in all experimental groups compared to 0 μg/mL (*p* < 0.05).

#### 3.3.3. CFU

The number of *S. mutans* colonies were counted to investigate the antibacterial activity of *Paeonia lactiflora* extract. The numbers of colonies at concentrations of 50 μg/mL, 100 μg/mL, 150 μg/mL, and 200 μg/mL were 21.3 ± 5.56, 16.0 ± 4.55, 15.0 ± 2.16 and 15.3 ± 4.86, respectively. For *C. albicans* the numbers of colonies at the same concentrations were 178.5 ± 9.95, 183.3 ± 9.39, 116.8 ± 7.89, and 111.3 ± 9.50, respectively (Figure 5).

#### 3.3.4. Morphology of Bacteria

The results of SEM also confirmed the antibacterial activity. The morphologies of microorganisms are shown in Figure 6. The cell membrane of bacteria was lost, and the streptococci chain was broken. And the shape of *C. albicans* changed irregularly compared to control group (0 μg/mL). In addition, control group for both bacteria *S. mutans* and *C. albicans* exhibited a larger number of cells that the 200 μg/mL group.

## 4. Discussion

In this study, the cytotoxicity and antibacterial activity of *Paeonia lactiflora* extract was investigated. The avoidance of synthetic additives is increasing as modern people demand safe and healthy drugs. Therefore, there is increasing demand for antibiotics made using natural extracts, and research to develop them is actively underway. Various studies using existing natural extracts have been conducted [17].

In our previous study, the antibacterial ability of the material containing the *Paeonia lactiflora* extract was confirmed, but the antibacterial activity of the *Paeonia lactiflora* extract undiluted solution was not confirmed [18].

The purpose of this study was to investigate *Paeonia lactiflora* extract as a possible antibiotic by verifying its composition, cytotoxicity, and antibacterial activity. In a previous study, the herb extracts were applied to *S. mutans* [19] and cacao beans were applied to streptococcus bacteria [20] to show antibacterial activity. When natural substances containing flavonoids and polyphenols were applied to *Helicobacter pylori*, they were shown to have an antibacterial effect [21]. The relationship between the concentration of flavonoids and polyphenols in *Paeonia lactiflora* extract and the antibacterial activity was investigated for each experiment group. In this study, the amounts of polyphenols and flavonoids increased as the concentration of the extract increased.

Therefore, the content of flavonoids and polyphenols in *Paeonia lactiflora* extract may be responsible for its antibacterial activity against *S. mutans* and *C. ablicans*. It is difficult to develop and apply antibiotics if it is cytotoxic, thus the cytotoxicity of *Paeonia lactiflora* extract at different concentrations was investigated. Therefore, in our study, four groups with different concentrations of flavonoids and polyphenols were selected, and cytotoxicity tests were performed first by extract concentration. To evaluate the stability of the extract, a cytotoxicity test was first applied. The cytotoxicity test [22] was performed with MTT-assay according to the iso regulation, and there was no significant difference in cytotoxicity between concentrations, and the survival rate was over 80%.

In addition to MTT assay, it did not show cytotoxicity when the extract was applied to evaluate the cell shape to confirm cytotoxicity. Antibacterial activity test and inhibition zone test were performed for antibacterial evaluation. The extract showed antibacterial activity. It showed antibacterial activity against *S. mutans* and *C. albicans*, the representative bacteria in the oral cavity.

Bacterial activity was lowered in the group to which the extract was applied. These results were like those of previous studies [23]. However, in previous studies, this extract was not applied to *S.mutans* and *C. albicans*, and significant results were obtained.

The CFU (colony forming units) decreased in the overall number of bacteria in the experimental group, compared to the control group. Additionally, the number of colonies of *S. mutans* bacteria was larger than the number of *C. albicans* bacteria. The fact that the number of colonies of bacteria is low with CFU, which is believed to suppress the ability of bacteria to grow, is an indicator of antibacterial activity. *S. mutans* had a lower survival rate than *C. albicans* when *Paeonia lactiflora* extract was used as an antibacterial agent

In addition, in the inhibition zone test, the control group did not show the bacterial death range, but the experimental group to which the extract was applied 1.20 ± 0.10, 1.33 ± 0.21, 1.40 ± 0.17, and 1.47 ± 0.15, respectively.

Since *Paeonia lactiflora* extract contains flavonoids, free radicals are generated, which have an antibacterial effect on bacteria. In particular, the *Paeonia lactiflora* extract was found to be effective against gram-positive bacteria, and in our study, it was effective against *S.muntans* and *C. albicans*. In addition, in previous studies, antibacterial effects were shown against B. subtilis, B, anthracis, S. aureus, and C. perfringens bacteria. [24].

Polyphenols and flavonoids identified in the extracts are confirmed to have antibacterial activity due to the generation of superoxide radicals, which can be linked to previous studies [24,25,26]. In particular, in our experiment, the bacteria to which the extract was applied showed antibacterial activity compared to the control group.

This result was consistent with the result of antibacterial activity. The results of CFU on the survival of bacteria also decreased as the concentration of the extract increased. These results served as an opportunity to prove the antibacterial power of the extract.

As a method to confirm the antibacterial activity of the extract against bacteria, SEM was applied to confirm the form of the bacteria to which the extract was applied. As a result of SEM, the appearance and shape of the bacteria were broken, and the chain of *S.mutans* was broken.

*C. albicans* was also confirmed that the shape of the bacteria was broken. These results were consistent with previous studies, and it was possible to prove the antibacterial activity of the extract.

In this experiment to which *Paeonia lactiflora* extract was applied, the extract did not show cytotoxicity and showed high bacterial activity. Therefore, it is confirmed that this extract can be used as an antibacterial agent for various oral bacteria in future studies as it has produced similar results to other extracts characterized by oral bacteria.

## 5. Conclusions

The *Paeonia lactiflora* extract was applied to *S. mutans* and *C. albicans* bacteria, the antibacterial activity of the natural extract could be confirmed compared to the control group. OD, CFU, SEM, and clear zone tests all showed antibacterial activity, and the MTT method was used to confirm the cytotoxicity of the extract.

For antibiotic substances to be safe for use inside the human body, they must not exhibit cytotoxicity. In this study, the cytotoxicity was low, making *Paeonia lactiflora* extract a candidate for use as an antibiotic.

More research should be conducted on the effects of *Paeonia lactiflora* extract on various bacteria and cells to confirm the level of cytotoxicity.

The conclusion of this study showed that the *Paeonia lactiflora* extract contained flavonoids and polyphenols, antibiotics that showed antibacterial activity effective against both *S. mutans* and *C. ablicans* bacteria.

## Figures and Tables

**Figure 1 medicina-58-01272-f001:**
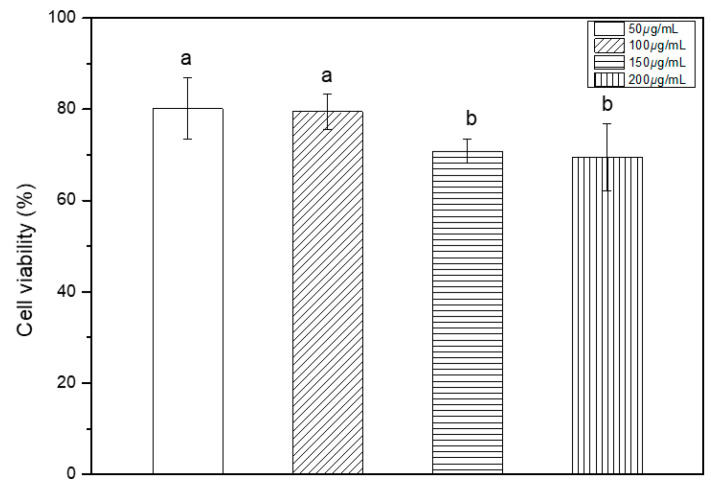
*Paeonia lactiflora* extract of cell viability (contents of 50 μg/mL, 100 μg/mL, 150 μg/mL, and 200 μg/mL.). ^ab^ The difference letter indicates statistics significant difference by One-Way ANOVA (*p* < 0.05).

**Figure 2 medicina-58-01272-f002:**
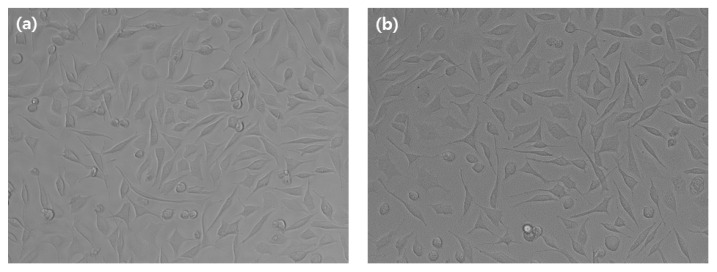
*Paeonia lactiflora* extract of cell morphology (**a**) control group (**b**) experimental group (200 μg/mL).

**Figure 3 medicina-58-01272-f003:**
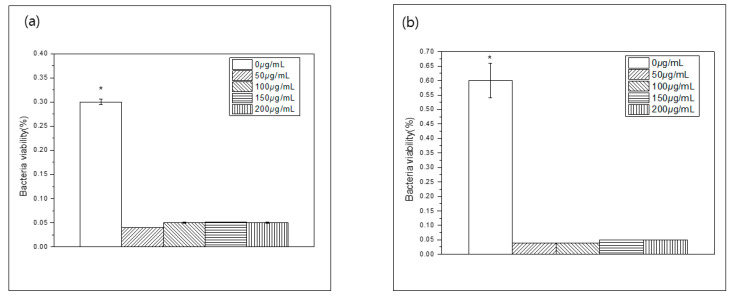
Result of bacteria viability test contents of 50 μg/mL, 100 μg/mL, 150 μg/mL, and 200 μg/mL (**a**) *S. mutans* group (**b**) *C. albicans* group. * The indicates statistics significant difference by One-Way ANOVA (*p* < 0.05).

**Figure 4 medicina-58-01272-f004:**
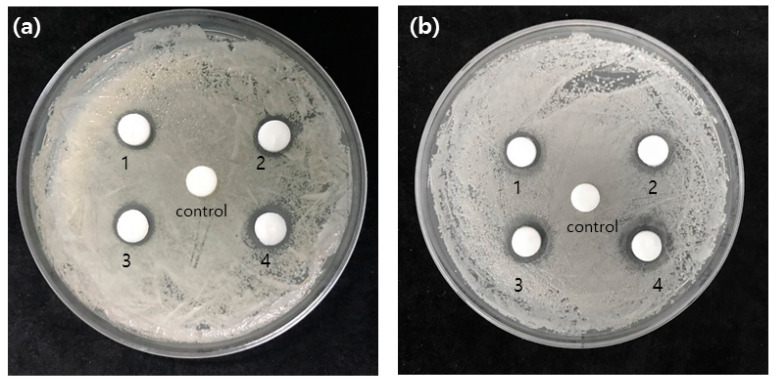
Inhibition zone test result (**a**) *S. mutans* group (1: 50 μg/mL, 2: 100 μg/mL, 3: 150 μg/mL, 4: 200 μg/mL) (**b**) *C. albicans* group (1: 50 μg/mL, 2: 100 μg/mL, 3: 150 μg/mL, 4: 200 μg/mL).

**Figure 5 medicina-58-01272-f005:**
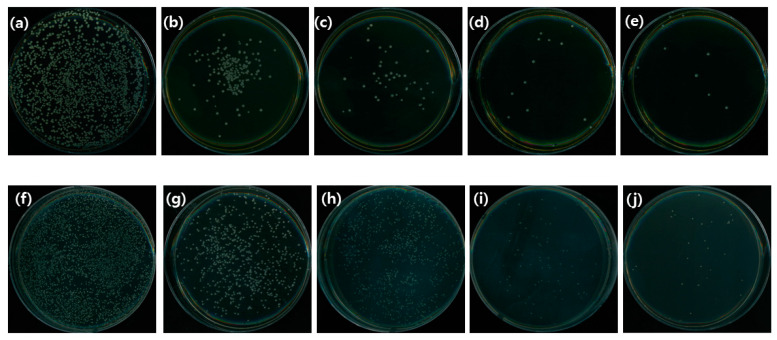
CFU results for *S. mutans* group (**a**) 0 μg/mL (**b**) 50 μg/mL (**c**) 100 μg/mL (**d**) 150 μg/mL (**e**) 200 μg/mL); *C. albicans* group (**f**) 0 μg/mL (**g**) 50 μg/mL (**h**) 100 μg/mL (**i**) 150 μg/mL (**j**) 200 μg/mL).

**Figure 6 medicina-58-01272-f006:**
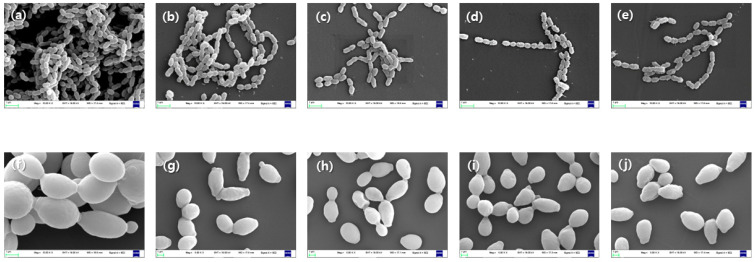
Result for bacterial morphology effect of *S. mutans control* group × 1000 (**a**) 0 μg/mL (**b**) 50 μg/mL (**c**) 100 μg/mL (**d**) 150 μg/mL (**e**) 200 μg/mL and *C. albicans* group: (**f**) 0 μg/mL (**g**) 50 μg/mL (**h**) 100 μg/mL (**i**) 150 μg/mL (**j**) 200 μg/mL.

**Table 1 medicina-58-01272-t001:** Concentration (µg/mL) of flavonoid and polyphenol in *Paeonia lactiflora* extract.

Type	Flavonoid	Polyphenol
0 μg/mL	0.049 ± 0.5 *	0.054 ± 1.1 *
50 μg/mL	18.6 ± 0.7	14.0 ± 2.1
100 μg/mL	19.1 ± 1.4	17.4 ± 3.0
150 μg/mL	19.6 ± 1.3	20.4 ± 4.6
200 μg/mL	18.5 ± 0.9	18.8 ± 2.3

* The indicates statistics significant difference by One-Way ANOVA (*p* < 0.05).

## Data Availability

Not applicable.

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
