# Peer review of "Evaluation of Cytotoxic and Antibacterial Effect of Methanolic Extract of Paeonia lactiflora"

_medicina, 2022, doi:10.3390/medicina58091272_

Round 1

Reviewer 1 Report

This manuscript is written about the antibacterial effects of Paeonia lactiflora extracts.

It will be better to revise some weaknesses.

Introduction

1. About Line 39. ‘To prevent and treat oral disease~’.

- Is it really necessary to control normal flora using antibacterial agents? 

- This introduction paragraph seems to mean that all oral bacterias should be eliminated(or controlled) even under normal conditions(not disease). 

- If you said to mean that, I don’t agree.

2. Introduction is a little meager.

- Please reinforce evidence about the necessity of antibacterial agents and natural extracts. It would give originality and necessity to your research.

- What is the active ingredient of the peony extract you estimated? - You should write more information about the active ingredients(you estimated..) from the peony extract. And use consistent term (for active ingredients).

Results.

- What does ‘a’ or ‘b’ mean in tables and figures? Information of abbreviations is insufficient. Provide a clear list of abbreviations used.

- Each table or figure is an independent message. It should contain all the information that will make the reader understand the message.

- Similar results seem to be too repetitive. (Of course, it’s different experiments)

Discussion.

- Do not repeat your results.

- You should discuss the meanings of results, not just results. - Describe in depth the differences from previous studies and the meaning of each result (the value of this study).

- The results of this experiment alone are insufficient for considering flavonoids and polyphenols as active ingredients of antibacterial effect. It is necessary to conduct additional experiments to supplement this, or at least conduct an investigation into the preceding literature. (It’s connected to the comment on the active ingredient of the introduction.)

Correct minor typo.

Author Response

Dear Reviewer

Thank you again for your comments, we appreciate them very much

Reviewer 2 Report

Article

Cytotoxicity and Antibacterial evaluation of neutral extract

This study is aimed to e investigated the antibacterial effect and the cytotoxicity of Paeonia lactiflora extrac

-          The study did not answered the research question efficiently

-          It is well written and designed

-          I have some comments should be considered for improving the article.

General points to be considered

v  Title: It is not clear and the name of the plant should be mentioned

Suggested titles: Evaluation of cytotoxic and antibacterial effect of methanolic extract of Paeonia lactiflora

v  In the current study, authors study the antibacterial effect on one type of bacteria (Streptococcus mutans) and this is not enough to report the antibacterial effects. Usually, we used 4-6 types of different types of gram +ve and gram -ve bacteria.

Abstract: looks good

Introduction: well written

Methods: need more illustrations

Results:

-          In table 1, why the authors add statistical analysis? It has no meaning and should be removed. In this table, assume we just present the values of flavonoids and polyphenols.

-          In figure 1 and 4, authors should write about the letters indicate for what? To compare between what????

-          . ab The difference letter indicates statistics significant difference by One-Way ANOVA (p<0.05).

-          Figure 3, shows that there is no any antibacterial effect as the size of the inhibited zones is very small indicating that there is no antibacterial effect of methanolic extract of Paeonia lactiflora.

 Discussion: needs to be expanded and should compare the results with previous studies.

v  Conclusion: must be corrected..

v  References: looks good, but still need to be checked following journal instructions.

Author Response

Dear Reviewer
Thank you again for your comments, we appreciate them very much
Please see the attachment

Reviewer 3 Report

Dear Authors,

I rated the article entitled:

Cytotoxicity and Antibacterial evaluation of neutral extract

Remarks:

1. the objectives of the study are unclear, please explain in the introduction what are the hypotheses from which you start, what are the objectives of the study.

2. (2.1.) - replace ~Mateiral~ with ~Material~.

3. what does it mean: Sancheong, Gyeongsangnam-do?! is it a company if yes, please specify the name of the company, city, country, etc.?!

4. the stages used in the study are unclear, respectively what objective was pursued. please reformulate the materials and methods chapter to understand what is to be analyzed.

5. in the results chapter, please present in detail the legends of the figures for a better understanding of the results.

6. please correlate the results with the discussions. Moreover, the discussions are written too briefly, please enter more details.

7. The conclusions are written too briefly.

Thank you

BR

Author Response

Thank you again for your comments, we appreciate them very much

Round 2

Reviewer 1 Report

Introduction

  • It would be recommended to add the content that “S.mutans bacteria are resident bacteria in the oral cavity, but when the amount increases, it causes dental caries.” (In addition to deleting a sentence from the previous manuscript.) 

Results

  • How many times have all the experimental results been measured? Are the values the means of the measurements? 
  • In particular, as flavonoids are considered important ingredients, I wonder how many times the content of the flavonoids according to the extract concentration has been measured. Please describe this in the manuscript.
  • I still can’t understand the meaning of a and b. I am not asking what this sentence in the legend means (The difference letter indicates statistics significant difference by One-Way ANOVA). What I’m talking about is the use of “a” and “b”. Significant differences mean the difference between the two groups (or more). A single column cannot be considered significant. Why is each column marked with significance? Do you mean “a” is the control? or.. If it means that there is a significant difference between “a” and “b”, how should we interpret figures with two “a” and “b”? Please see figures or tables in other papers.
  • “Each table or figure is an independent message. It should contain all the information that will make the reader understand the message.” This means that each table or figure is independently described from the text of the manuscript. The information should be provided even if it is already described in the text. Please describe the meaning of “error bars” or abbreviations, etc.

Discussion

  • Paragraph 3 in Discussion.. It is still unclear. The explanation that flavonoids and polyphenols are the active ingredients of Paeonia lactiflora extract does not seem reasonable. Why are flavonoids and polyphenols estimated to be active ingredients in antibacterial agents? Is it because only “herb extract and cacao beans had antibacterial effects” and “natural substances containing flavonoids and polyphenols showed antibacterial effect”? Is this the reason for presuming flavonoids and polyphenols as the active ingredients of antibacterial agents? Evidence-based explanations for this causal relationship are required.
  •  You said that “the amounts of polyphenols and flavonoids increased as the concentration of the extract increased.” However, concentrations decreased in 200 μg/ mL (Table 1). What does this mean? Does it mean anything? Does this difference not affect the antibacterial effect of the extract? 

Correct typos.

Author Response

Dear reviewer.

Thank you for your valuable comments.

Revisions to the thesis are attached in this file.

Reviewer 2 Report

It looks good now.

Author Response

Dear reviewer.

Thank you very much for your advice.

sincerely yours.

Reviewer 3 Report

Dear Authors,

The paper looks better.

BR

Author Response

(The authors gave the same response as above.)

Round 3

Reviewer 1 Report

The manuscript has been properly revised.

Please check for typos.